# Clifford Algebraic Rotor Embeddings : Maybe embeddings should start to CARE

**Sameeksha Sriram**[1,*], **Timo Lüddecke**[1], **Ayush Paliwal**[1,2],
**Alexander S. Ecker**[1,2], **Chase van de Geijn**[1,†]
[1]Institute of Computer Science and Campus Institute Data Science, University of Göttingen
[2]Max Planck Institute for Dynamics and Self-Organization, Göttingen, Germany
[*]sameeksha.sriram@stud.uni-goettingen.de
[†]chase.geijn@uni-goettingen.de

## Abstract

Rotary Positional Embeddings (RoPE) have demonstrated exceptional performance as a positional encoding method, consistently outperforming their baselines. While recent work has sought to extend RoPE to higher-dimensional inputs, many such extensions are non-commutative, thereby forfeiting RoPE's shift-equivariance property. Spherical RoPE is one such non-commutative variant, motivated by the idea of rotating embedding vectors on spheres rather than circles. However, spherical rotations are inherently non-commutative, making the choice of rotation sequence ambiguous. In this work, we explore a quaternion-based approach—Quaternion Rotary Embeddings (QuatRo)—in place of Euler angles, leveraging quaternions' ability to represent 3D rotations to parameterize the axes of rotation. We show Mixed RoPE and Spherical RoPE to be special cases of QuatRo.

Further, we propose a generalization of QuatRo to *Clifford Algebraic Rotary Embeddings* (CARE) using geometric algebra. Viewing quaternions as the even subalgebra of $Cl(3, 0, 0)$, we extend the notion of rotary embeddings from quaternions to Clifford rotors acting on multivectors. This formulation enables two key generalizations: (1) extending rotary embeddings to arbitrary dimensions, and (2) encoding positional information in multivectors of multiple grades, not just vectors. We present preliminary experiments comparing spherical, quaternion, and Clifford-based rotary embeddings.

## 1 Introduction

Rotary positional embeddings (RoPE) have proven remarkably effective in language modeling, prompting interest in adapting their success to higher-dimensional domains such as vision and multimodal learning [Siméoni et al., 2025]. Extending RoPE beyond 1D sequences requires careful consideration to preserve properties such as relative positional dependence (shift-equivariance) and reversibility [Liu and Zhou, 2025, Su, 2021]. While early extensions sought to preserve strict equivariance [Yu et al., 2025, Schenck et al., 2025], recent findings suggest this property may not be essential for strong performance, opening the door to non-commutative generalizations [van de Geijn et al.].

Spherical RoPE [van de Geijn et al.] exemplifies this approach: the rotations are performed on a sphere — a space where rotations fail to commute — thus breaking strict equivariance. Its implementation uses Euler angles (yaw and roll matrices), which explicitly constrain the rotations to be around the principal axes. Rather utilizing the Euler angles, one could parameterize rotations with quaternions, an idea mused in Su [2021], but discarded due to their non-commutativity. This allows for simple parameterization of the axes of rotation.

39th Conference on Neural Information Processing Systems (NeurIPS 2025) Workshop: UniReps 2025.

In this work, we revisit Quaternion Rotary Embeddings (QuatRo). Quaternions offer a compact, stable representation of 3D rotations allowing us to parameterize the axes of rotation rather than the assumed principal axes of Spherical RoPE. We further generalize QuatRo to *Clifford Algebraic Rotary Embeddings* (CARE), leveraging the geometric algebra framework. By interpreting quaternion rotors as grade-2 blades of $Cl(3, 0, 0)$ acting on grade-1 vectors, we derive a principled method for:

1. Generalizing rotary embeddings to arbitrary dimensions.
2. Allowing embeddings to inhabit multivector spaces, enabling richer positional transformations across grades.

This generalization not only subsumes quaternion-based methods but also creates new possibilities for encoding positional structure in higher-dimensional and multimodal settings. While CARE generalizes QuatRo to higher-dimensional data such as video or point clouds, this work is still in progress, and experiments are currently restricted to 2D images.

## 2 Related Work

Several recent efforts have sought to extend Rotary Positional Embeddings (RoPE) to higher-dimensional data [Su et al., 2024]. The most general formulation to date is *LieRE* [Ostmeier et al., 2024], which models RoPE as a rotation of $D$-dimensional query sub-vectors via the exponential of a linear combination of skew-symmetric matrices. Under the standard proof of RoPE's relative positional property, these generators must commute, as can be seen through the Baker–Campbell–Hausdorff formula. This constraint has led prior work to impose commutativity requirements on the rotation generators to preserve strict shift-equivariance [Yu et al., 2025, Schenck et al., 2025, Liu and Zhou, 2025].

However, recent studies have questioned the necessity of these constraints. In particular, van de Geijn et al. propose *Spherical RoPE*, which applies rotary encodings on the sphere—a setting where rotations do not commute—showing that performance can remain competitive despite breaking equivariance. Their approach parameterizes rotations using Euler angles, introducing potential issues such as gimbal lock and unintuitive composition behavior.

Our method, *Quaternion Rotary Embeddings* (QuatRo), builds on this line of work by replacing Euler angles with quaternion rotations. Quaternions can be viewed as a compact and numerically stable representation of 3D rotations, corresponding to the even subalgebra of $Cl(3, 0, 0)$. While QuatRo can be interpreted as a special case of LieRE with fixed $3 \times 3$ skew-symmetric generators, our generalization—*Clifford Algebraic Rotary Embeddings* (CARE)—extends beyond LieRE's scope. CARE treats rotary embeddings as Clifford rotors acting on multivectors, enabling both higher-dimensional generalization and graded (multi-grade) positional encoding.

The relationship between LieRE and CARE is nuanced: in one view, LieRE can be seen as a restricted subclass of CARE where the generators are limited to certain skew-symmetric matrices; in another view, certain CARE configurations reduce to LieRE.

## 3 Background

Due to space constraints, we focus on the specific algebraic tools and notation relevant to our method, and refer the reader to Roelfs and Keninck [2021] for comprehensive dives into geometric algebra and rotors and van de Geijn et al. for $N$-D positional encodings.

### 3.1 Quaternions and Rotors

Quaternions form a four-dimensional non-commutative algebra over the real numbers, with basis $\{1, \mathbf{i}, \mathbf{j}, \mathbf{k}\}$ and multiplication rules:

$$\mathbf{i}^2 = \mathbf{j}^2 = \mathbf{k}^2 = \mathbf{ijk} = -1,$$
$$\mathbf{ij} = \mathbf{k}, \quad \mathbf{jk} = \mathbf{i}, \quad \mathbf{ki} = \mathbf{j},$$

with anti-commutativity for distinct basis elements, e.g., $\mathbf{ji} = -\mathbf{ij}$. A quaternion $q = a_0 + a_{\mathbf{i}}\mathbf{i} + a_{\mathbf{j}}\mathbf{j} + a_{\mathbf{k}}\mathbf{k}$ can be split into a scalar part $a_0$ and a vector part $\mathbf{v} = a_{\mathbf{i}}\mathbf{i} + a_{\mathbf{j}}\mathbf{j} + a_{\mathbf{k}}\mathbf{k}$.

Pure quaternions (zero scalar part) can represent 3D vectors, while *unit quaternions* represent 3D rotations. Given a unit quaternion rotor

$$r = e^{\frac{\theta}{2}\mathbf{u}} = \cos(\theta/2) + \mathbf{u}\sin(\theta/2),$$

where $\mathbf{u}$ is a unit pure quaternion indicating the rotation axis, a vector $\mathbf{a}$ is rotated via

$$\mathbf{a}' = r\,\mathbf{a}\,r^{-1}.$$

This provides smooth composition of rotations as they can be composed with the quaternion product $r = r_1 r_2$.

### 3.2 Geometric Algebra and Clifford Rotors

Quaternions are isomorphic to the even subalgebra of $Cl(3,0,0)$, making them a special case of Clifford rotors. This perspective allows generalization to higher dimensions by replacing quaternion rotors with the bivectors of $Cl(n,0,0)$. These generalized rotors perform rotations with the operation,

$$\mathbf{a}' = e^{\frac{\theta}{2}\mathbf{B}}\,\mathbf{a}\,e^{-\frac{\theta}{2}\mathbf{B}}.$$

where $e^{\frac{\theta}{2}\mathbf{B}} = \cos(\theta/2) + \mathbf{B}\sin(\theta/2)$. $\mathbf{B}$ is a unit bivector which defines the

| Quaternions | | | | | Even-Grade CA | | | |
|---|---|---|---|---|---|---|---|---|
| | $1$ | $i$ | $j$ | $k$ | | $1$ | $e_{12}$ | $e_{23}$ | $e_{13}$ |
| $1$ | $1$ | $i$ | $j$ | $k$ | $1$ | $1$ | $e_{12}$ | $e_{23}$ | $e_{13}$ |
| $i$ | $i$ | $-1$ | $k$ | $-j$ | $e_{12}$ | $e_{12}$ | $-1$ | $e_{13}$ | $-e_{23}$ |
| $j$ | $j$ | $-k$ | $-1$ | $i$ | $e_{23}$ | $e_{23}$ | $-e_{13}$ | $-1$ | $e_{12}$ |
| $k$ | $k$ | $j$ | $-i$ | $-1$ | $e_{13}$ | $e_{13}$ | $e_{23}$ | $-e_{12}$ | $-1$ |

Figure 1: Multiplication tables of quaternions and the even subalgebra of $Cl(3,0,0)$, illustrating their isomorphism.

plane of rotation, analogous to the unit quaternion. However, in contrast to quaternions, Clifford rotors can act not only on vectors (grade-1) but also on higher-grade elements, enabling a potentially richer positional encoding schemes.

### 3.3 Rotary Positional Embeddings (RoPE)

RoPE encodes absolute positions by applying a position-dependent rotation to query and key vectors in the attention mechanism. For the $i$th 2D sub-vector of the query and key $(x_i, y_i)$, RoPE applies a rotation by angle $\theta_p$ determined by the position $p$:

$$\mathbf{q}_i' = \begin{bmatrix} x_i' \\ y_i' \end{bmatrix} = \begin{bmatrix} \cos\theta_i p & -\sin\theta_i p \\ \sin\theta_i p & \cos\theta_i p \end{bmatrix} \begin{bmatrix} x_i \\ y_i \end{bmatrix}.$$

In this formulation, $p$ is one dimensional, however it can be extended to 2D inputs. One method of extending RoPE to 2D is to perform two spherical rotations to 3D sub-vectors of the query and key.

$$\mathbf{q}_i' = \begin{bmatrix} x_i' \\ y_i' \\ z_i' \end{bmatrix} = \begin{bmatrix} 1 & 0 & 0 \\ 0 & \cos\theta_i p_1 & -\sin\theta_i p_1 \\ 0 & \sin\theta_i p_1 & \cos\theta_i p_1 \end{bmatrix} \begin{bmatrix} \cos\theta_i p_2 & -\sin\theta_i p_2 & 0 \\ \sin\theta_i p_2 & \cos\theta_i p_2 & 0 \\ 0 & 0 & 1 \end{bmatrix} \begin{bmatrix} x_i \\ y_i \\ z_i \end{bmatrix}.$$

## 4 QuatRo

While Spherical RoPE rotates around two of the principle axes, we propose using quaternion algebras, Quaternion Rotary (QuatRo) embeddings, to easily compose arbirary rotations. This allows us to parameterize the axes of rotation in each sub-vector rather than rotating around orthogonal axes. To do this, we parameterize $\theta^{(x)} = [a_i^{(x)}, a_j^{(x)}, a_k^{(x)}]$ such that,

$$\mathbf{r}_x = a_i^{(x)}\,\boldsymbol{i} + a_j^{(x)}\,\boldsymbol{j} + a_k^{(x)}\,\boldsymbol{k}, \qquad r_x = e^{\mathbf{r}_x p_x},$$

and similarly for $p_y$. We then apply the rotations to the query or key sub-vector.

$$\mathbf{q}_i' = r_x r_y\,\mathbf{q}_i\,r_y^{-1} r_x^{-1}$$

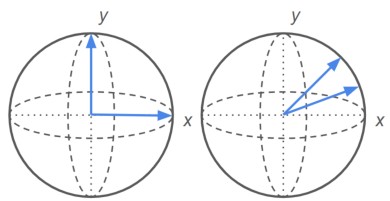

Figure 2: QuatRo can rotate around any two axes unlike Spherical RoPE.

Table 1: Performance comparison (top-1 accuracy) on CIFAR100 across methods and implementations.

| Fixed Encoding | Top-1 Accuracy (%) on CIFAR100 | | | |
| --- | --- | --- | --- | --- |
| | **Mixed** | **Spherical** | **QuatRo** | **CARE** |
| Quaternion Framework | 74.3 | 74.2 | 74.3 | |
| Clifford Algebraic Framework | 74.8 | 74.0 | 74.1 | 74.8 |
| ViT Baselines | **Absolute PE** | 64.2 | **Axial RoPE** | 72.1 |

The difference between QuatRo and Spherical RoPE is illustrated in Fig 2. While QuatRo is equivalent to Spherical RoPE when the axes are orthogonal, QuatRo is equivalent to Mixed RoPE when the axes of rotation are parallel. By fixing the axis of rotation and only learning the rotation speeds, we can implement both Mixed and Spherical RoPE in the context of QuatRo. To keep the emphasis on whether generalization is worth while and reduce the effects of "lucky" implementations, we compare the accuracies of each method implemented within the QuatRo framework.

## 5  CARE

We generalize QuatRo by allowing the Clifford rotors act on all grades of the multivector. Rather than 3D sub-vectors, the query and key are split into 8D sub-vectors corresponding to each scalar coefficient in Cl(3,0,0). However, the rotor is represented by only 3 parameters per dimension for each position coordinate, like in QuatRo.

As QuatRo can be seen as a case of CARE where the sub-vector of the query/key is restricted to the grade-1 (vector) component, we can once implement all methods within the CARE framework. In Table 1, we show the accuracy of these methods in both frameworks trained with a ViT-B [Dosovitskiy et al., 2020] on CIFAR100.

## 6  Discussion

QuatRo generalizes both Spherical and Mixed RoPE, so we would expect it to have higher performance. However, we see that among all, CARE performs the best along with Mixed RoPE implemented in the clifford algebraic framework. Mixed RoPE's performance over Spherical RoPE suggests that there is a small benefit to strict equivariance as an inductive bias. However, this may also be noise from better random initialization.

While conceptually CARE generalizes the other methods, in practice, it is more restrictive since it requires 8 coefficients per sub-vector. However, our initial experiments indicate that CARE preforms better than Spherical or Quaternion RoPE. While it would be interesting to look its utilization of each grade, we have not done this as of yet. We predict that it may emphasize the scalar channels which remain position invariant.

**Future Work**   We plan to use CARE to expand to higher dimensional data such as 3D point clouds. We also intend to experiment with different algebras. For example, one could perform rotations in space-time algebra which allows for rotations in hyperbolic space.

**Limitations**   In its current form, QuatRo is marginally slower than previous methods, however CARE is significantly slower due to the geometric product. It is unclear if this limit is unsurpassable or due to the current lack of well optimized Clifford Algebra computations. Additionally, conclusions are hard to draw from the results of one run on CIFAR100.

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

## A  CARE formulation

Let $q_i \in \mathbb{R}^8$ encode the coefficients of a multivector in the $\mathrm{Cl}(3,0)$ basis corresponding to $(1, e_1, e_2, e_{12}, e_3, e_{31}, e_{23}, e_{123})$. For a position $p = (p_x, p_y)$, we define per-axis bivectors $B_x, B_y \in \wedge^2 \mathbb{R}^3$ and angles $\theta_x(p_x), \theta_y(p_y)$. The CARE rotor is applied via the sandwich product:

$$\tilde{q}_i = R_y(p_y)\, R_x(p_x)\, q_i\, R_x(p_x)^{-1}\, R_y(p_y)^{-1}, \quad R_\alpha(p_\alpha) = \exp\!\Big(\tfrac{1}{2}\, \theta_\alpha(p_\alpha)\, B_\alpha\Big),$$

with $\alpha \in \{x, y\}$. We parameterize each $B_\alpha$ by a unit 3D axis (three degrees of freedom, shared with QuatRo), and each $\theta_\alpha(\cdot)$ by standard RoPE frequency schedules. This preserves QuatRo's parameter efficiency while acting on all grades of the multivector via a single rotor field.

Crucially, this framework subsumes previous variants: vector-only CARE with orthogonal bivectors recovers *Spherical RoPE*, even-grade CARE restricted to the even subalgebra reduces to *QuatRo*, and parallel-axis CARE collapses to *Mixed RoPE*.

