# OpenReview forum: "Clifford Algebraic Rotor Embeddings : Maybe embeddings should start to CARE"
_NeurIPS.cc/2025/Workshop/UniReps — UniReps2025_

### Official Review · Reviewer_nmhc · 2025-09-12
**Clifford Algebraic Rotor Embeddings:Maybe embedding should start to CARE**

**Confidence:** 3

**Review:**

Quality
The study introduces Quaternion Rotary Embeddings (QuatRo) and their generalisation to Clifford Algebraic Rotary Embeddings (CARE). By interpreting quaternions as the even subalgebra of Cl(3,0,0), the authors extend rotary positional embeddings (RoPE) to act on vectors and multivectors across multiple grades, thereby enabling richer positional transformations in arbitrary dimensions. The technical foundations are solid. Quaternions and Clifford rotors are rigorously defined. CARE’s formulation via bivectors and rotor sandwiches is mathematically consistent, and CARE is shown to subsume Spherical RoPE, Mixed RoPE, and QuatRo as exceptional cases.

Clarity
The exposition is generally clear and well structured, with a good flow from RoPE → QuatRo → CARE.
Some sections (especially the CARE formulation in the appendix) are dense and algebra-heavy; more intuition and geometric illustrations would benefit non-specialists.
The title and abstract effectively convey novelty but could better highlight the practical implications (for example, when CARE should be expected to help).

Originality
The paper is highly original, and extending positional encodings into Clifford algebraic frameworks is novel.
CARE goes beyond LieRE (Lie algebra–based RoPE generalisation) by allowing multigrade encodings and arbitrary-dimensional generalisations.
Combining Mixed RoPE, Spherical RoPE, and QuatRo under CARE provides conceptual clarity and a general mathematical lens.

Significance

Scientific significance: CARE provides a unifying and extensible mathematical framework for positional embeddings, potentially crucial for multimodal and high-dimensional data.

Practical significance: Currently limited. CIFAR100 experiments show only marginal gains; CARE is computationally slower due to geometric product overhead. The authors acknowledge limitations and suggest optimised CA libraries and further datasets.

Future impact: If extended to larger models and modalities, CARE could establish a new line of algebraically principled positional embeddings connecting symmetry, equivariance, and representation alignment, central to UniRep themes.


Pros
Novel and mathematically elegant extension of RoPE into Clifford algebra.
The CARE framework subsumes prior variants (Mixed, Spherical, and QuatRo), offering conceptual unification.
Preliminary empirical validation shows CARE is competitive.
Opens new research directions (higher-dimensional encodings, space-time algebra, multimodal embeddings).

Cons
Experiments are limited (CIFAR100, ViT-B, one run).
Mixed RoPE outperforms CARE in practice, raising questions about when the added algebraic generality helps.
CARE is slower computationally, limiting scalability.
Some derivations are dense and not easily interpretable for readers without a background in Clifford algebras.

**Score:**

4

**Topic Fit:**

2

---

### Official Review · Reviewer_D2hT · 2025-09-13
**This paper introduces Clifford Algebraic Rotary Embeddings (CARE), a generalization of rotary positional embeddings (RoPE) to arbitrary dimensions using Clifford algebra. The authors start by revisiting Quaternion Rotary Embeddings (QuatRo) as an alternative to Spherical RoPE, showing that QuatRo subsumes both Mixed and Spherical RoPE by allowing arbitrary axes of rotation parameterization. They then generalize further to CARE by treating quaternions as elements of the even subalgebra of  Cl(3,0,0) Cl(3,0,0) and using Clifford rotors to act on multivectors of multiple grades. Preliminary experiments on CIFAR-100 with ViT-B suggest CARE performs slightly better than Spherical and Quaternion RoPE, but conclusions are limited due to the small-scale experiments. The work is positioned as a step toward richer positional encodings for higher-dimensional and multimodal tasks.**

**Confidence:** 4

**Review:**

Overall Evaluation

The paper proposes a theoretically motivated generalization of RoPE by framing positional embeddings in the language of geometric algebra. This is an important contribution as it unifies multiple prior approaches (Mixed RoPE, Spherical RoPE, QuatRo) under a single formalism and opens up new possibilities for richer, multi-grade positional encodings. However, the empirical evaluation is relatively limited, leaving open questions about practical benefits in large-scale settings.

Strengths (Pros):

1. Originality & Theoretical Rigor

Introduces CARE, a principled and mathematically elegant generalization of rotary embeddings to arbitrary dimensions using Clifford algebra.

Provides a unifying framework where Mixed RoPE, Spherical RoPE, and QuatRo emerge as special cases.

Clear mathematical exposition of quaternions, Clifford rotors, and their connection to positional encoding.

Highlights potential for future extensions to point clouds, video, and multimodal settings.

2. Clarity

Well-structured with clear sections: Introduction, Background, QuatRo, CARE, Discussion.

Good use of algebraic notation and formulas.

Multiplication tables (Figure 1) and comparisons (Table 1) are informative and aid understanding.

3. Significance

CARE provides a conceptual bridge between Lie group-based approaches (LieRE) and more general geometric algebra frameworks.

Potentially impactful for higher-dimensional data representations (vision, multimodal, spatial reasoning).


Weaknesses (Cons):

1. Empirical Evaluation is Limited

Experiments are restricted to CIFAR-100 with ViT-B, which may not generalize to large-scale vision or multimodal tasks.

No ablation studies to analyze the effect of using different grades in CARE (e.g., scalar vs. bivector channels).

Limited discussion of computational overhead — CARE is noted to be slower, but no quantitative benchmarks are given.

2. Practical Impact Unclear

While theoretically elegant, it is unclear whether CARE leads to significant performance gains over simpler methods (e.g., Mixed RoPE).

Results show only marginal improvements, which could be within noise levels.

3. Clarity

Some terminology (e.g., "rotor field", "multi-grade embeddings") could be better explained for readers unfamiliar with geometric algebra.

The paper might benefit from a more explicit discussion of why non-commutativity may be beneficial in practice.

Overall Impression:

This paper is theoretically strong and novel, but empirically underdeveloped. It would benefit from a more extensive experimental section, including large-scale datasets and deeper analysis of performance trade-offs. Nonetheless, it lays a solid foundation for future research on algebraically grounded positional encodings.

**Score:**

3

**Topic Fit:**

3

---

### Official Review · Reviewer_S4vU · 2025-09-16
**CARE provides a way to extend RoPE to higher-dimensional inputs while maintaining shift-invariance**

**Confidence:** 2

**Review:**

## Motivation

The paper seeks to extend RoPE to higher-dimensional inputs while maintaining shift-invarance, addressing the limitations of methods like Spherical RoPE, which suffers from being non-commutative.

## Main Results

- The paper formulates QuatRo, and CARE as a generalization of QuatRo, showing that CARE can generalize rotary embeddings to arbitrary dimensions, enabling potentially richer positional transformations

- Using top-1 accuracy on CIFAR100 as the lone practical evaluation criterion, the authors show that CARE embeddings slightly outperform Spherical RoPE, but not Mixed RoPE.

## Strengths

- Clear motivation and problem formulation: the theoretical shortcomings of RoPE and spherical RoPE are clear, whether or not they extend to practical implications for deep neural nets.

- Novel theoretical framework using Clifford algebra to extend RoPE to higher dimensions and generalize rotary embeddings in general.

## Concerns & Questions

- A single practical evaluation criterion was used, which did not show evidence that CARE is practically useful. This is a major limitation. Until shown otherwise, not sure why we should start to CARE?

- Why not apply to additional model architectures with additional metrics?

- The authors mention that CARE is "significantly slower" than other rotary embedding methods. How much slower? Some benchmarks for this would be nice.

## Overall Impressions

- CARE seems like a theoretically elegant framework for generalizing rotary embeddings using Clifford algebra. The motivation is clear, but the results on the single evaluation and the practical computational considerations are significant shortcomings.

- Even if CARE proves not to be practical for positional embeddings for contemporary AI systems, this negative result could still be helpful to know for the field, if additional experiments and analysis and evals are done.

**Score:**

2

**Topic Fit:**

2